# Liquid Nuclear Magnetic Resonance (NMR) Spectroscopy in Transition—From Structure Elucidation to Multi-Analysis Method

Thomas Kuballa [1,*], Katja H. Kaltenbach [1,2], Jan Teipel [1] and Dirk W. Lachenmeier [1]

[1] Chemisches und Veterinäruntersuchungsamt (CVUA) Karlsruhe, Weissenburger Strasse 3, 76187 Karlsruhe, Germany; jan.teipel@cvuaka.bwl.de (J.T.); lachenmeier@web.de (D.W.L.)

[2] Max Rubner-Institut, Nationales Referenzzentrum für Authentische Lebensmittel, Haid-und-Neu-Strasse 9, 76131 Karlsruhe, Germany

[*] Correspondence: thomas.kuballa@cvuaka.bwl.de; Tel.: +49-721-926-3639

**Abstract:** As early as 1946, Felix Bloch and Edward Mills Purcell detected nuclear magnetic resonance signals, earning themselves the Nobel Prize in 1952. The same year saw the launch of the first commercial nuclear magnetic resonance (NMR) spectrometer. Since then, NMR has experienced significant progress in various fields of application. While in the 1970s NMR spectroscopy was solely employed for determining the structure and purity of synthesis products in the chemical field, it gradually gained popularity in the medical field for the investigation and rendering of images of human organs. Since then, the technique has developed significantly in terms of stability, reproducibility, and sensitivity, thereby forming the foundation for high-resolution imaging, the automation or standardization of analytical procedures, and the application of chemometric methods, particularly in relation to identifying food adulteration. This review objectively assesses the current state of implementing liquid NMR in the food, cosmetics, and pharmaceutical industries. Liquid NMR has transitioned from a structural elucidation tool to a widely recognized, multi-analytical method that incorporates multivariate techniques. The illustrations and sources provided aim to enhance novice readers' understanding of this topic.

**Keywords:** nuclear magnetic resonance (NMR) spectroscopy; food; cosmetics; pharmaceuticals; structure determination; purity; chemometrics; food adulteration

## 1. Introduction

In 1952, Felix Bloch and Edward Mills Purcells were awarded the Nobel Prize for their discovery of nuclear magnetic resonance signals in 1946 [1–3]. During the same year, Varian Associates from Palo Alto, CA, USA released the first commercial nuclear magnetic resonance spectrometer. Additionally, Bruker, founded by GÜNTHER LAUKIEN in Karlsruhe, Germany, built the first commercial nuclear magnetic resonance pulse spectrometers. In the subsequent years, nuclear magnetic resonance (NMR) spectroscopy has rapidly advanced in various application fields. Where NMR spectroscopy was primarily utilized for determining the structure and purity of synthesis products exclusively in the chemical industry in the 1970s, it has since grown in popularity within the medical industry for exploring and imaging human organs.

Through technological advancements, particularly in metrological implements and computer technology, NMR has evolved significantly in terms of stability, reproducibility, and sensitivity. This has formed the foundation for high-resolution imaging, automated processes, and the standardization of analytical procedures. The spectrum of NMR applications encompasses, nowadays, the identification and structure elucidation of organic and biochemical molecules, the quantitative detection of single or multiple analytes, and non-targeted screening when paired with chemometric methods. Notably, there is a rising

trend in the development of methods focusing on food adulteration, including origin determination, production methods, and the verification of declared contents. This indicates that NMR spectroscopy is one of the most flexible current analytical techniques [4].

NMR is a primary method of measurement according to metrology standards [5]. It has been proven to have good reproducibility. NMR spectra are evaluated numerically, including chemical shifts, coupling constants, and signal integrals, which lead to quantitative results. For the targeted evaluation of NMR spectra, i.e., the quantification of known components in a sample, the intensities of different signals recorded under properly adjusted and identical experimental conditions are directly proportional to the number of nuclei giving rise to their respective resonance signals. There are three possible approaches to quantitative NMR: adding an internal standard (ISTD) of known concentration to the analysis sample [6]; comparing the analysis sample to another sample with an external standard (PULCON method—Pulse-Length-Based Concentration Determination [7]); or sending a calibrated, synthetic signal to be recorded with the analyte sample spectrum that mimics the signal of an internal standard. (ERETIC method—Electronic Reference To access In vivo Concentrations [8,9]). See Section 4 for more details. In view of these facts, it is peculiar that the term NMR *spectroscopy* is still commonly used despite being inconsistent with linguistics as well as with IUPAC definitions: in Ancient Greek, $\sigma\kappa o\pi\acute{\epsilon}\omega$ (skopéō) means "to see" and $\mu\acute{\epsilon}\tau\rho\acute{\epsilon}\omega$ (métréō) means "to measure". The IUPAC Gold Book defines spectroscopy and spectrometry as follows: spectroscopy is the study of physical systems by the electromagnetic radiation with which they interact or that they produce and spectrometry is the measurement of such radiations as a means of obtaining information about the systems and their components [10]. Maybe chemists should "promote" NMR from spectroscopy to spectrometry, minding the quality of NMR data and avoiding contradictory terms. Despite these considerations, the orthodox term spectroscopy is used in this publication.

NMR is utilized in food surveillance and quality control and several articles have highlighted the potential of this technique before [11–14]. Additionally, [2]H NMR spectroscopy—site-specific natural isotope fractionation (SNIF) as another NMR technique—has been proven to be an effective tool in wine analytics, regarding geographical origin, since the 1990s [15]. Methods for [1]H NMR in food surveillance have been published only in the last two decades. Monitoring also raises questions about the determination of origin and authenticity. In addition to established methods, such as isotope mass spectrometry, NMR spectroscopy is also a promising approach to authenticate and assess the quality of food and cosmetics. In 2003, LE GALL and COLQUHOUN [4] summarized the fundamental suitability of NMR for origin determination. Owing to the numerous analysis options in this field, development is progressing rapidly. Thus, samples can be subjected to identifications, structure elucidations, and quantifications, as well as authenticity and origin determinations based on their spectra. A further advantage is that, under defined conditions, it is possible to retrospectively evaluate past samples analyzed since all spectral information is stored and can be accessed. Numerous possibilities have been reported in the literature, many of which demonstrate the use of chemometric methods.

Since all spectral information is recorded in a core-specific manner, it is possible to quantify ingredients, make comparisons, discriminate, or classify, for example, when evaluating the authenticity of foodstuffs or determining the origin and variety of specific products. Furthermore, non-target analysis allows a fast and highly selective sample screening with significant information gain. Key features of the NMR technique are its often low sample preparation requirement and acceptable measurement time, enabling a high sample throughput. The NMR spectra of, e.g., food products, usually contain numerous signals and, therefore, are highly informative. This may appear initially disadvantageous for classical spectral evaluation. However, chemometric techniques, such as multivariate data analysis, can aid in the visualization and assessment of the data.

Chemometrics involves the use of mathematical and statistical methods in chemistry. This discipline of formal logic can be used to plan experimental designs or evaluate experi-

mental measurement data [16]. For analyses of multivariate data, different methods could be used, e.g., principal components analysis, cluster analysis, and multiple linear regression. The fundamental concept of many chemometric techniques rests on the application of latent variables, e.g., principal components. These variables are depicted in the space of the original data to simplify intricate and voluminous data sets and uncover any obscure dependencies [17]. In addition, using this technique, analytical measurement data can be easily graphically portrayed and interpreted. The initial analysis of NMR spectra with chemometrics was presented in 1971 by KOWALSKI et al. [18]. Over the years, chemometrics became increasingly popular, partly due to more powerful computers. Subsequently, publications featured methods such as linear discriminant analysis [19] and the classification of objects into specific groups, which is referred to as the Soft Independent Modelling of Class Analogy (SIMCA) [20]. These methods can be found in numerous publications and textbooks [21–32].

In this review, practical examples are provided to demonstrate how high-resolution NMR is applicable in the quality control and surveillance of food, cosmetics, and pharmaceuticals.

## 2. Practical Aspects of NMR Application

### 2.1. Sample Preparation

The examination of food, cosmetics, and pharmaceuticals using NMR typically requires little sample preparation. This review focuses on liquid-state NMR. A liquid sample for NMR analysis should be a one-phase, non-turbid solution (otherwise signal broadening can result). Filtration or centrifugation can yield clear measurement samples. The weighed portion of the extracted or dissolved original sample material should be precisely known. For quantification with an external standard, the total volume of the sample (original material plus solvent) should be precisely known. If extraction is used to prepare the sample, a solvent ensuring the optimal and reproducible solubility of the target analytes should be selected. Systematic underdetermination, e.g., due to incomplete but well-reproducible extraction, can be compensated with correction factors ascertained during method validation.

High-throughput analyses for general quality monitoring are preferably conducted using sample extracts or solutions. Due to the price of the miniature rotors (the sample vessels) used in MAS-NMR (magic angle spinning, solid-state NMR), this is rather uneconomic for high throughput samples; thus, MAS-NMR is used only for specific analyses of solid or viscous samples (e.g., oncological biopsies, gel studies) if the sample should not or cannot be dissolved.

For instance, fats and oils are usually weighed, blended with deuterated chloroform (including tetramethylsilane as an internal standard), and measured directly. For the quantitative or chemometric analysis of samples containing labile/sensitive substances, the usage of mitigated chloroform (thoroughly stripped of phosgene and hydrogen chloride traces) is advised here to prevent the deterioration of target substances and/or skewed spectra due to pH-induced irregular chemical shifts [33].

Aqueous matrices, such as soft drinks, fruit juices, or milk, are mixed with $D_2O$ (including trimethylsilyl propionic acid as an internal standard)—if necessary after pH adjustment—before direct measurement [34,35]. Ethanol-containing drinks, such as beer, spirits, or wine, can be quantified by adding a buffer and, if required, making a pH adjustment [36–38]. For solid food, cosmetics, or mixtures of pharmaceuticals, extractions (aqueous or with organic solvents, depending on the analytical purpose) are needed prior to the preparation of the measurement solution. One significant advantage of NMR analyses over other methods is that they often do not necessitate the complicated isolation and purification of individual substances during sample preparation. High-field NMR instruments, which are widely used, offer high resolution (half-width of a signal compared to spectral width), enabling the assessment of complex mixtures by univariate and multivariate analysis, even with minimal sample preparation.

### 2.2. NMR Spectroscopy

The NMR spectra of fat-containing matrices in solvents such as $CDCl_3$ can be recorded directly. However, in the $^1H$ NMR spectroscopy of matrices containing water or ethanol, the $H_2O$ protons or the protons of the methyl or methylene group of ethanol dominate the spectrum to an extent that makes electronic signal amplification for the purpose of detecting minor components impossible. However, these signals may be effectively attenuated through appropriate presaturation, allowing the detection of trace-level constituents via electronic amplification [36,39,40]. This approach can be employed to uncover adulteration within a given spectrum (e.g., the use of methanol to adulterate spirits). Furthermore, collecting spectra in a consistent manner is critical for automated assessments and chemometric analysis. Among other things, it is important to ensure accurate phasing, an accurate baseline, a consistent smoothing factor, a constant signal half-width, and the chemical shift of pH-sensitive components. This required reproducibility of spectral data is a fundamental prerequisite for mathematical operations in chemometrics. The great amount of information and data density found in high-resolution NMR spectra often presents an issue, even for current computer systems; therefore, data reduction is frequently performed. A common approach is the binning or bucketing method [36]. This involves dividing the spectra into small segments, usually of constant width, and determining the total signal area of each segment. The resulting matrix of integrals can be processed via chemometric procedures. For $^1H$ NMR spectra, segments of 0.01 to 0.05 ppm are typically useful for static bucketing. It is advisable to exclude regions of solvent signals.

Protium ($^1H$) is the ideal candidate for NMR due to its near 100% natural abundance, its high gyromagnetic ratio of 42.6 MHz/T, and its diverse presence in organic molecules. Being a spin $\frac{1}{2}$ nucleus, protium yields very sharp resonance signals compared to nuclei with a higher spin. Other than $^1H$, other spin $\frac{1}{2}$ nuclides, such as $^{13}C$, $^{15}N$, $^{19}F$, and $^{31}P$, are potential candidates for qNMR but certain factors are limiting their usefulness: $^{13}C$ has a natural abundance of only 1.1% and a gyromagnetic ratio of 10.7 MHz/T, the resulting severely lower detection efficiency compared to $^1H$ necessitates higher concentrations or a high number of accumulated scans. On the other hand, $^{13}C$ NMR spectra show signals for every carbon, which is useful for analyte identification. If broadband $^{13}C$ {$^1H$} decoupling is used, the $^{13}C$ signal intensities are not quantitative due to the potential nuclear Overhauser effect. Additionally, $^{15}N$ has a natural abundance of only 0.36% and a gyromagnetic ratio of only $-4.32$ MHz/T, again, leading to a severely low detection efficiency compared to $^1H$. The abundance (~100%) of $^{19}F$ and its gyromagnetic ratio (40.1 MHz/T) allow for the fast acquisition of highly sensitive spectra and fluorine's chemical shift range is very wide, compared to the shift range of $^1H$; thus, even with mostly only a few fluorine atoms in a target molecule, the specific shift is a strong aid for identification. With an isotopic abundance of ~100% and a rather high gyromagnetic ratio (17.2 MHz/T), $^{31}P$ is also a useful nuclide for routine NMR, especially focusing on bioorganic molecules and certain complexes. For more detailed aspects of hetero-nuclei NMR, see [41–43].

A key aim of method development should be to ensure properly set and equal experimental conditions (acquisition parameters), yielding optimal spectra and good comparability between spectra (e.g., for quantification with an external standard). There are important parameters to optimize for quantitative accuracy. A sufficient delay time for full relaxation (of the evaluated nuclei) between subsequent excitation pulses is ensured, avoiding specific correction factors. A spectral width encompassing approx. 3 ppm of clean baseline to both sides of the region with signals is ensured, easing better phase and baseline correction. An excitation frequency is set centrally in the region with signals, improving overall equal excitation. Optimized excitation pulses (preferably 90° pulses) of known duration and power (especially for aqueous samples with varying electrolyte concentrations) are important for comparability between sample spectra and external standards (PULCON method). Overly strong signals from protonated solvents or water can be attenuated by signal suppression; then, (weaker) signals relevant for evaluation can be acquired better.

NMR spectra intended for quantification or chemometric evaluation generally benefit from the following processing steps: a zero filling doubling the number of real data points, a light exponential line broadening (<0.5 · FWHM, the full width at half maximum of a clean resonance) to improve the signal-to-noise ratio, and $0^{th}$-order phase correction resulting in pure absorption mode signals; spectra needing a $1^{st}$-order phase correction should be re-acquired with better parameters because $1^{st}$-order phasing often leads to hard-to-adjust baseline distortions. The baseline correction should ensure a flat baseline (for empty regions), with the median of the noise at zero.

In many cases, simple quantification is possible by evaluating a 1D spectrum; the spectra of complex mixtures can be evaluated by precisely identifying the relevant signals with a JRES (J-resolved 2D NMR spectra) first and, then, using the 1D spectrum for quantification. If signals are superimposed, a guided curve fit (deconvolution) based on prior knowledge about the target signal will lead to improved quantitative accuracy.

For more details, see the cited literature, e.g., [40,44–46].

## 3. Structure Elucidation and Confirmation

Numerous publications detailing various pulse sequences and experiments have been published in the scientific literature over recent decades, e.g., those outlined by BERGER and SICKER [47,48] JACOBSEN [49], FRIEBOLIN [50], and KEELER [51], to aid in structure elucidation and confirmations using NMR techniques.

Other than one-dimensional NMR experiments, such as $^{1}$H, $^{13}$C, or further hetero-nuclei experiments ($^{15}$N, $^{19}$F, $^{31}$P), two-dimensional homo- and hetero-nuclear experiments, such as COSY (correlation spectroscopy), TOCSY (total correlated spectroscopy), INADE-QUATE (Incredible Natural Abundance DoublE QUAn-tum Transfer Experiment), JRES, HSQC (heteronuclear single quantum coherence), HMBC (heteronuclear multiple bond correlation), and other experiments, could be conducted and combined.

For the verification of the identity of active pharmaceutical ingredients, various methods with differing levels of significance can be employed. These methods include melting point determinations, as well as UV/VIS and infrared spectroscopy, and HPLC or GC combined with UV or mass spectrometry. Nonetheless, it can be problematic to fully establish the identity and purity of active pharmaceutical ingredients in practice. Even when these methods are used in combination, they may not provide unambiguous verifications of identity.

With NMR spectroscopy, analysts have a tool that often provides a rapid method for identifying the active ingredient in a drug and detecting possible impurities. Especially in the development of synthesis processes, quality control, and the surveillance of drugs on the market, an easy-to-perform one-dimensional $^{1}$H NMR spectrum often fulfills all requirements. To determine whether signals from impurities overlay with the main signals of the known target ingredients, the whole $^{1}$H NMR spectrum should be carefully scanned to detect additional minor signals. If the evaluation of some of the signals of the target analyte does not result in the same degree of purity, partial deterioration of the target analyte or signal overlay should be considered. However, to ensure the purity of a sample in complex cases, 2D-NMR experiments are recommended. By a comparison of the found target analyte concentration (determined from the NMR spectrum) with the weighed portion, the target analyte's purity can be determined, even in the presence of NMR-inactive impurities or hygroscopically adsorbed water.

The complex structure of 2-butyl-4-chlor-1-{4-[2-(1H-tetrazol-5-yl)phenyl]benzyl}imidazole-5-methanol, well known as an antihypertensive agent with the trade name losartan, is a suitable example. Losartan can be confirmed through a simple $^{1}$H NMR experiment, measured under the conditions described by ACKERMANN et al. [52]. The signals in the measured NMR spectrum, as shown in Figure 1, correlate well with predicted chemical shifts. Since no other signals appear in the NMR spectrum other than the pertinent signals, it is possible to confirm the identity of the active ingredient and rule out impurities within the standard NMR-detection limits in the lower ppm range. In rare instances, it may

be suitable to utilize nitrogen NMR, [13]C NMR, and, if required, two-dimensional homo- and heteronuclear experiments or DFT (density functional theory) calculation methods to calculate shielding constants for confirmation purposes.

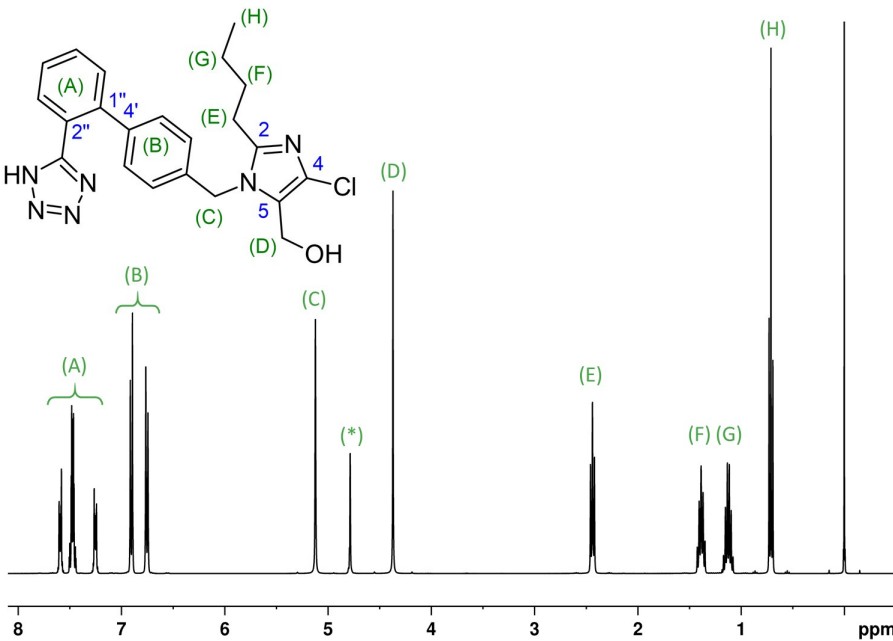

**Figure 1.** The [1]H NMR spectrum (400 MHz) of losartan (potassium salt, $C_{22}H_{23}ClN_6O$), 16.07 mg/mL in $H_2O/D_2O$; acquisition with water suppression and referenced to $\delta_{TSP}$ = 0.00 ppm (TSP is trimethylsilyl propionic acid); upper side: the structure with signal assignments; lower side: the measured spectrum. Losartan-K [1]H NMR ($H_2O/D_2O$ 9:1, ISTD: TSP, 400 MHz, 300 K): $\delta_H$ 7.62–7.56 (m, 1H), 7.51–7.43 (m, 2H), 7.29–7.22 (m, 1H), 6.93–6.87 (dt, 2H, *J* = 8.2 Hz), 6.78–6.72 (dt, 2H, *J* = 8.2 Hz), 5.121 (s, 2H), [4.78 water (suppressed)], 4.370 (s, 2H), 2.483 (t, 2H, *J* = 7.5 Hz), 1.388 (quin, 2H, *J* = 7.5 Hz), 1.124( sx, 2H, *J* = 7.4 Hz), 0.710 (t, 2H, *J* = 7.4 Hz) Note:* the signal at 4.78 is the remnant of the suppressed solvent signal; due to proton exchange the resonance of NH is typically broad and weak and, thus, also not really visible. The hydroxyl proton gives no resonance, due to the fast exchange with solvent. The unambiguous assignment of the aromatic protons (A) and (B) can be affirmed with two dimensional NMR spectra.

## 4. Quantification and Purity Check for Quality Management

NMR spectroscopy is widely accepted as a primary method, with its analysis quantity being directly proportional to a physical quantity, such as the signal area in the spectrum. This fact makes it inherently quantitative and suitable for quantitative analysis [53] while keeping the instrument parameters and criteria of ISO IEC 17025 [54] in mind for officially recognized laboratories. Many analytical practices, such as external or internal calibration or standard addition, can be utilized. A significant benefit of the NMR technique is the potential for quantification without the requirement of an analyte reference substance and via a separate certified reference standard. Vice versa, the reliability of NMR makes it a useful tool for the preparation of reference materials [55] and current research focuses on enhancing metrological NMR's potential further [13].

The three possible approaches to quantitative NMR in more detail:

Firstly, a precisely known amount of internal standard (ISTD) [6] can be dissolved in the analysis sample; the ISTD shall be chosen so its signal(s) do not overlap with signals from the original sample.

If the observed nuclei of compounds in the sample are fully relaxed before excitation, their resonance signal intensities are directly proportional to the amount of substance in

the sample. Formula (1) allows the determination of a target analyte's concentration in a sample by comparison with the exactly known concentration of an internal standard:

$$\frac{I_{Q_j}}{I_{An_i}} = \frac{N_{Q_j}}{N_{An_i}} \cdot \frac{m_{ISTD}}{m_S} \cdot \frac{M_{An}}{M_Q} \cdot \frac{P_Q}{P_S} \tag{1}$$

where:

An: refers to the analyte in the sample;
Q: refers to the qNMR (quantitative NMR) standard in the internal standard;
S: refers to the sample;
ISTD: refers to the internal standard;
$I_{An}$: is the integrated signal area of the analyte in the sample (A);
$I_Q$: is the integrated signal area of the qNMR standard (Q);
$N_{An}$: is the number of resonating protons causing the analyte's signal that is integrated;
$N_Q$: is the number of resonating protons causing the qNMR standard's signal that is integrated;
$m_S$: is the mass (weighed portion) of the sample (S);
$m_{ISTD}$: is the mass (weighed portion) of the internal standard (IS);
$M_{An}$: is the molar mass of the analyte in the sample (A);
$M_Q$: is the molar mass of the qNMR standard (Q);
$P_S$: is the purity (mass fraction) of the sample (S);
$P_Q$: is the purity (mass fraction) of the qNMR standard (Q) in the ISTD;
subscript *i*: refers to the *i*-th signal from the analyte in the sample (A);
subscript *j*: refers to the *j*-th signal from the qNMR standard (Q) in the ISTD.

To ensure full relaxation, the relaxation pause $T_r$ between accumulated scans fulfills $T_r > 5 \cdot T1_i$ (the longest T1 relaxation value, typically under 4 s, seldom over 10 s).

Secondly, an external standard solution containing (a) precisely known amount/s of at least one reference substance/s can be measured using the same experimental conditions before or after the analysis sample (PULCON method [7]). Formula (2) calculates the response (sensitivity) of the spectrometer, yielding the PULCON factor:

$$f_{PULCON} = \frac{I_{Ref} \cdot SW_{Ref} \cdot M_{Ref}}{SI_{Ref} \cdot \gamma_{Ref} \cdot N_{H,Ref}} \left( \text{in} \frac{\text{a.u.} \cdot \text{ppm} \cdot \text{L}}{\text{mol}} \right) \tag{2}$$

where:

$f_{PULCON}$: is the PULCON factor;
$I_{Ref}$: is the absolute integral of the reference signal;
$SW_{Ref}$: is the spectral width (e.g., 20.55 ppm);
$M_{Ref}$: is the molar mass of the reference standard;
$SI_{Ref}$: is the no. of data points of the processed reference spectrum (e.g., 131,072 = 128 Ki = $2^{17}$);
$\gamma_{Ref}$: is the purity-adjusted mass concentration of the reference substance;
$N_{H,Ref}$: is the number of protons per reference molecule causing the evaluated resonance.

The PULCON factor of a current, well-maintained NMR spectrometer is usually stable for days to weeks; even so, for quality control, a daily measurement and recalibration is sensible.

Once this PULCON factor is known, it can be used to calculate concentrations of analytes in other samples using Formula (3):

$$\gamma_{An} = \frac{I_{An} \cdot SW_{An} \cdot M_{An} \cdot V_{solv} \cdot k_{An}}{SI_{An} \cdot f_{PULCON} \cdot N_{H,An} \cdot m_{An} \cdot f_{dil}} \cdot \frac{P_{An}}{P_{Ref}} \cdot \frac{NS_{Ref}}{NS_{An}} \tag{3}$$

where:

$\gamma_{An}$: is the analyte mass concentration (in mg/kg);

$I_{An}$: is the absolute integral of the analyte in the sample;

$SW_{An}$: is the spectral width (e.g., 20.55 ppm);

$M_{An}$: is the molar weight of the analyte;

$V_{solv}$: is the volume of (deuterated) solvent used in the extraction;

$k_{An}$: is the correction factor (if needed, e.g., isomeric ratio, recovery ratio under the used experimental conditions);

$SI_{An}$: is the number of data points of the processed analyte spectrum (e.g., 131,072 = 128 ki = $2^{17}$);

$f_{PULCON}$: is the mean value PULCON factor from QuantRef;

$N_{H,An}$: is the number of protons per analyte molecule giving this resonance;

$m_{An}$: is the weighed portion of the original sample material used in the dissolution or extraction (in kg);

$f_{dil}$: is the dilution factor from analyte stock solution to measurement sample, e.g., 0.8, if 20% (*v/v*) buffer was added;

$P_{An}$: is the excitation pulse length used for the analyte sample;

$P_{Ref}$: is the excitation pulse length used for the QuantRef solution;

$NS_{Ref}$: is the number of accumulated scans for the reference spectrum;

$NS_{An}$: is the number of accumulated scans for the analyte spectrum.

If the excitation pulse lengths and/or number of scans are the same between the analysis sample and reference, they reduce to one and may be left out of the equation.

Thirdly, an artificial signal with an intensity calibrated against a known concentration of a reference substance can be generated, appearing in an otherwise empty spectral region (ERETIC method [8,9]). This signal is recorded with the analysis sample spectrum and used as if it were an internal standard signal.

Usually, samples for [1]H NMR experiments are dissolved in deuterated solvents to prevent the intense signal(s) of the solvent's protons from drowning out the weaker signals of the analyte(s). For many life-sciences-related samples (e.g., clinical samples, food, feed), the exclusion of water is an obstacle; however, due to the well-defined resonance frequencies, such unwanted signals can be suppressed precisely [45]. The application of solvent suppression is a routine allowing for faster sample preparation and saving expensive deuterated solvents.

For samples that are complex mixtures with many signals over the whole chemical shift range, it can be a challenge to find an appropriate internal standard. In such cases, the concentrations of individual compounds can be ascertained using the PULCON principle (Pulse-Length-Based Concentration Determination) [7,56]. It has been established that one measurement can quantify a vast range of different analytes in fruit juice, wine, alcoholic spirits, and soft drinks [46,52,57]. Thus, ACKERMANN et al. [52] validated and established a standard procedure for the quantification of several analytes in non-alcoholic beverages. Based on this accredited in-house procedure, an analysis protocol was included in the German Official Collection of Methods of Analysis in 2021-11: "Determination of several Ingredients, Additives and Contaminants in alcohol-free Refreshing Beverages with Quantitative Nuclear Resonance Spectrometry" [58]. The major constituents and trace components of these drinks can be detected using [1]H NMR spectroscopy down to the lower mg/L concentrations. Detection at even lower concentrations is theoretically possible after enrichment; however, this raises questions about the required effort and better suitability of alternative detection methods. Examples of analytes in non-alcoholic beverages after the protocol of ACKERMANN et al. [52] are shown in the [1]H NMR spectra in Figure 2.

It is not uncommon for food samples to contain several different carbohydrates, leading to a region densely packed with resonance signals between ~4.5 ppm and ~3.2 ppm (see Figure 2). A potential approach to allow the determination of compounds, even in such situations, is the systematic derivatization of the sugars for the purpose of reducing overlapping signals, see, e.g., [59].

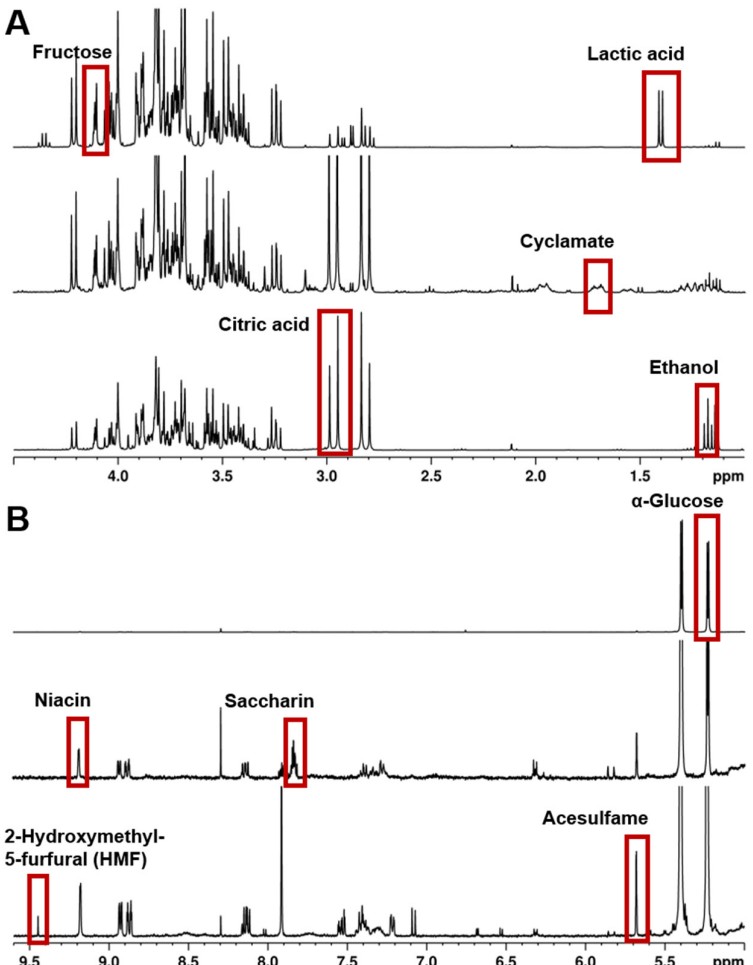

**Figure 2.** The [1]H NMR spectra (400 MHz) in $H_2O/D_2O$, acquisition with water suppression and referenced to $\delta_{TSP} = 0.00$ ppm, of three different non-alcoholic beverages (top: lemonade, middle: diet lemonade, bottom: energy drink (syrup)). (**A**) Low field region 1.0–4.5 ppm and (**B**) high field region 4.5–9.6 ppm. Examples of analytes for the quantification are highlighted.

The NMR technique can also clearly detect adulterated and illegal alcohol. For example, 304 spirits were examined, revealing alcohols with health-relevant contents, such as diethyl phthalate or polyhexamethylene guanidine in Russian samples [60] and ethyl carbamate and methanol in Romanian samples [61].

Using the example of the so-called free aldehydes rancidity parameter in fats and cosmetics, the quantitative suitability of NMR can be further demonstrated. Lachenmeier et al. [62] have already provided ample description of the study's findings. Higher aldehydes are crucial indicators of autoxidative fat spoilage, referred to as so-called rancidity. Despite sensory analysis, no standard method is available to objectively assess these findings as the gas chromatographic methods designed for oils did not prove effective for cosmetic matrices. Edible oils and fats can be directly measured after being dissolved in a solvent while lipsticks require extraction with deuterated chloroform. With [1]H NMR, the detection of alkanals is accomplished through the identification of a triplet in the low field at around 9.8 ppm. Additionally, frequently added flavoring substances in cosmetics can also be determined through the [1]H NMR measurement. Signal overlaps may be present only between 7-hydroxycitronellal and alkanals and they may, therefore, require validation by GC-MS. The method was completely validated [62]. Another example related to consumer health protection is the determination of mineral oil saturated hydrocarbons (MOSH) and mineral oil aromatic hydrocarbons (MOAH) in cosmetics and cosmetic raw materials [63]. For the NMR measurements, samples need to be diluted in deuterated chloroform; a limit

of detection of <0.1 g/100 g was validated. Compared with other analytical techniques (liquid chromatography or gas chromatography), the NMR method stands out due to its higher specificity [63].

The general quality, reliability, and efficiency of NMR analyses have also led to its increasing application in pharmaceutical research and analysis [64–66] and in physiological chemistry and metabolism research [67,68].

In principle, multivariate calibration using the PLS method can provide quantifications [28]. Models for calculating original gravity, ethanol, and lactic acid were established by LACHENMEIER et al. [36] through the PLS method, which correlated NMR spectra with results from reference methods. MONAKHOVA et al. [69] used the PLS method to analyze parameters including methanol, higher alcohols, 2-phenyl alcohol, and ethyl acetate in spirits. The study found a substantial correlation with the gas chromatographic reference analysis, as well as with ethyl carbamate [70]. MARTÍNEZ-SABATER et al. [71] applied various chemometric techniques, among them PLS, to assess alterations in organic matter throughout the composting of newly generated winery and distillery waste by means of the main spectral ranges present during the process.

The evolution of pulse sequences has accompanied the progress of NMR in various fields, predominantly for qNMR. These sequences comprise a series of radio frequency pulses that are radiated onto the samples in NMR tubes located in the sample head. If energy is irradiated as a radio frequency pulse corresponding to the Larmor frequency of the nucleus of interest, the resonance condition is satisfied and transitions between different states are possible. In a coordinate system rotating around the *z*-axis with axes x′ and y′, the radio frequency pulse's irradiation deviates the macroscopic magnetization by a particular angle θ concerning the *z*-axis. A 90° radio frequency pulse deflects the magnetization M0 into the x′y′ plane, resulting in transverse magnetization My′. The mangling vector then precesses around the *z*-axis at the Larmor frequency, generating an oscillating voltage in the detector coil. The resulting electrical signals from the coil are known as the free induction decay (FID) and can be acquired. Following a Fourier transformation, the typical NMR spectra are obtained. There are numerous publications and textbooks available in the literature for the fundamental implementation of pulses and pulse sequences. Notably, the works of BODENHAUSEN et al. [72] and the textbooks of BERNSTEIN et al. [73] and LEVITT [74] should be referenced. The initial application of NMR in food science was in 1957, with an attempt to evaluate moisture in food using low-resolution NMR [75]. Abbreviations for technical terms should be explained thoroughly upon first mention. For applications involving complex matrices in aqueous or aqueous/ethanolic matrices, commonly present in food items, more intricate pulse sequences with single or multiple water suppression are required, using high-resolution NMR [40,76,77]. The high stability of all acquisition parameters over extended periods enables obtaining accurate quantification despite multiple suppressions, particularly when utilizing the PULCON principle [40]. Nonetheless, when conducting suppression experiments, precautions should be taken to guarantee that signals in close proximity are suppressed proportionally, depending on the degree of irradiance. Through validation studies, any under-results can be statistically recorded and compensated for by correction factors upon the evaluation of key figures. This ensures that quantitative determinations do not incur systematic underestimations. The analysis of proteins relies heavily on frequently used two-dimensional experiments, which largely stem from the pioneering work of KURT WÜTHRICH, a Nobel Prize laureate in 2002 [78].

## 5. Chemometric Methods (Multivariate Data Analysis)—Food Fraud and Authenticity Assessment

According to the European Commission, olive oil, milk, honey, saffron, orange juice, apple juice, grape wine, vanilla extract, and fish are the most common sources of food fraud in the European Union [79]. The authentication and identification of adulteration in foodstuffs, pharmaceuticals, or cosmetics can be achieved by using appropriate databases

with the aid of [1]H or [13]C NMR in combination with chemometric methods. Generally, all the data collected from the database are compared with the specific core information and visually displayed. In the chemometric classification approach, the data reduction of the original data is often undertaken using so-called unsupervised methods, such as PCA (principal components analysis) or ICA (independent components analysis), and combined with a so-called supervised method [80]. In practice, supervised methods, such as decision tree methods, linear or quadratic discriminant methods, support vector methods (linear or quadratic), the k-nearest neighbor method, or probabilistic classifiers based on naive Bayes, are commonly used. For deeper general information on chemometrics or multivariate data analysis, see, e.g., [75,81–83].

Multivariate NMR methods have frequently been cited in the literature as a dependable tool for detecting instances of food fraud and pharmaceuticals, particularly in recent years as the prevalence of counterfeit products has become a growing concern over the past decade [84–87]. FLÜGGE et al. [88] and PACHOLCZYK-SIENICKA et al. [89] demonstrated the use of the method on spices, SCHMITT et al. [90] on walnuts, and MONAKHOVA et al. [91] on medicinal products. The following examples are intended to demonstrate the application.

*5.1. Wine*

The literature contains several descriptions of [1]H NMR wine analysis [80,92,93]. For example, a sum of 281 samples of wine was prepared and measured, as described by MONAKHOVA et al. [80]. For the PCA-LDA (principal components analysis followed by linear discriminant analysis) analysis, raw data were scaled to the specific spectrometer ERETIC factor; aligned to 3-(trimethylsilyl)propionic acid (−0.2–0.2 ppm); bucketed with 600 buckets (between 0.5 and 9.2 ppm), using exclusions for water and ethanol (0.95–1.355; 3.6–3.7; 4.6–6.8 ppm); and logarithmized with the Factor 5. The PCA was conducted with 19 dimensions, no autoscaling, and a confidence of 95%. As demonstrated in Figure 3, a highly accurate classification was attained through cross-validation using PCA-LDA of the [1]H NMR spectra from three grape cultivars: Cabernet Sauvignon (47 samples), Tempranillo (90 samples), and Pinot Noir (144 samples). The cross-validation was a randomized 10-fold procedure, using 90% of the samples for building the model and 10% of the samples for the verification of assignment. After comparing other supervised methods, such as decision trees, quadratic discriminant, support vector methods (linear or quadratic), the k-nearest neighbor method, and naive Bayes, it was found that all supervised classification methods yield similar acceptable results. There was also no significant difference found between linear discriminant analysis and support vector methods (linear or quadratic) [94].

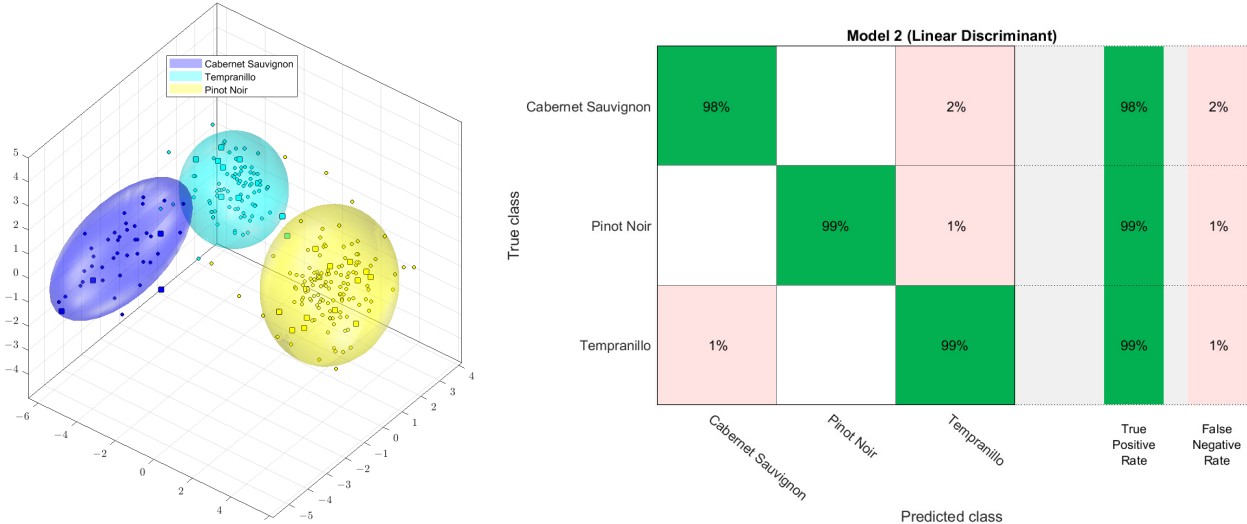

**Figure 3.** PCA−LDA score plot (**left**) and confusion matrix (**right**) of the [1]H NMR spectra of three wine grape varieties: Cabernet Sauvignon, Tempranillo, and Pinot Noir.

### 5.2. Edible Oils

Proton NMR investigations on edible oils have been reported in the literature using both targeted and non-targeted techniques. SKIERA et al. utilized [1]H NMR to quantify free fatty acids [95], peroxide value [96], and anisidine value [97]. Furthermore, a range of edible oils, including soybean, sunflower, canola, olive, and coconut oil, can easily undergo examination [98]. Alternatively, MAESTRELLO et al. [99] employed metabolomics methods to verify the specific quality trait of extra virgin olive oil.

A quick assessment of edible oils can be achieved through the PCA-LDA of the [1]H NMR spectra. For this example, fourteen samples of walnut oil, ninety-two samples of olive oil, thirty-five samples of pumpkin seed oil, six samples of peanut oil, eight samples of linseed oil, and eight samples of rapeseed oil, all purchased from local markets, were prepared by dissolving 200 mg of edible oil in 800 μL of deuterated chloroform, which includes 0.1% tetramethylsilane as an internal standard, and measuring under the conditions described by MONAKHOVA et al. [100]. For the PCA-LDA, the raw data were scaled to glycerine at 4.1 to 4.2 ppm, aligned to TMS ($-0.2$–0.2 ppm), bucketed with 800 buckets (between 0.5 and 9.2 ppm), used with exclusions for solvent (6.32–6.39; 7.1–7.4; 7.65–8.02 ppm), and logarithmized with the Factor 1. The PCA was conducted with 40 dimensions, no autoscaling, and a confidence of 99.99%.

Since the complete (fat) metabolome is utilized in the non-targeted analysis, both adulteration and spoilage can be promptly identified. Figure 4 displays a Monte Carlo cross-validation by means of a score plot and the percentage of correct assignments. The Monte Carlo cross-validation was a randomized 10-fold procedure, using 90% of the samples for building the model and 10% of the samples for the verification of assignment. Due to differences in fatty acid profiles, edible oils can be classified effectively. The method, which involves relatively simple sample preparation, is suitable for identifying unusual oils for further investigation in routine analyses.

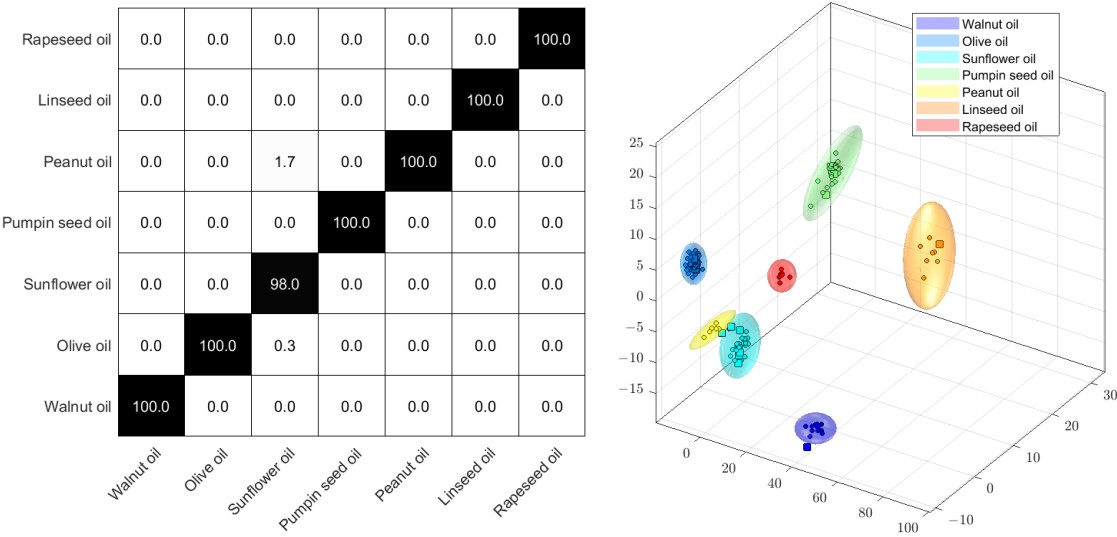

**Figure 4.** PCA−LDA confusion matrix (**left**) and scores plot (**right**) of the [1]H NMR spectra of different edible oils.

### 5.3. Honey

Honey is a natural food and it is worth noting that bee honey has been one of the most commonly adulterated foods [79]. There are analytical approaches for the determination of its botanical and geographical origin through various analytical methods, such as sensory analysis, color, pollen analysis, electrical conductivity, and the measurement of the concentration of main and minor sugars. However, sophisticated adulterations are often difficult to detect by the named classical methods. For this reason, quality control with reliable methods is essential. NMR is capable of capturing the entire chemical metabolome

of honey and provides sufficient resolution, making it well-suited for both quantitative analyses and chemometric evaluations.

The analytical authenticity testing and quality testing of honey involve quantifying its contents, including sugars; organic acids; quality indicators, such as ethanol; or heating markers, like hydroxymethylfurfural [101]. These are then compared to authentic reference samples to ascertain their quality. Moreover, various studies have demonstrated the effectiveness of utilizing NMR to trace the origin of honey [102,103]. Zielinski et al. demonstrated the use of NMR and chemometrics techniques for identifying the botanical source of Polish monofloral honeys through NMR spectroscopy [104].

The classification of the botanical origin can easily be conducted by the application of PCA-LDA to [1]H NMR data. For this example, the honey samples of three botanical origins were prepared and measured under conditions described by Gerhardt et al. [105]. For the PCA-LDA, the raw data of 30 acacia honeys, 32 rape honeys, and 33 forest honeys were scaled to sugar area (3.1 to 4.15 ppm); aligned to glucose at 4.6274 ppm; bucketed with 600 buckets (between 0.5 and 9.2 ppm), using exclusions for water (4.70–4.95 ppm); and logarithmized with the Factor 1. The PCA was conducted with 25 dimensions, no autoscaling, and a confidence of 99%. The Monte Carlo cross-validation was a randomized 10-fold procedure, using 90% of the samples for building the model and 10% of the samples for the verification of assignment.

Figure 5 displays the Monte Carlo cross-validation by means of a percentage of correct assignments (confusion matrix) and the score plot of the PCA-LDA. It is common knowledge that the strongest and most predominant signals in the [1]H NMR spectra of honey stem from the sugar regions between 3.0 and 5.5 ppm, usually alpha- and beta-glucose and fructose [105]. However, logarithmization decreases the significance of these sugars as primary components for multivariate analysis in preference of minor constituents, like organic carboxylic acids or amino acids, together with various other compounds, such as formic acid, ethanol, or hydroxymethylfurfural. Since these minor components demonstrate noticeable intensity variations between honey specimens of varying botanical origin, this method also facilitates excellent classification. As illustrated in the confusion matrix depicted in Figure 4, it is possible to almost correctly classify all samples, with 100% accuracy for rapeseed and acacia honey and 99.4% for forest honey, even with the relatively small sample group.

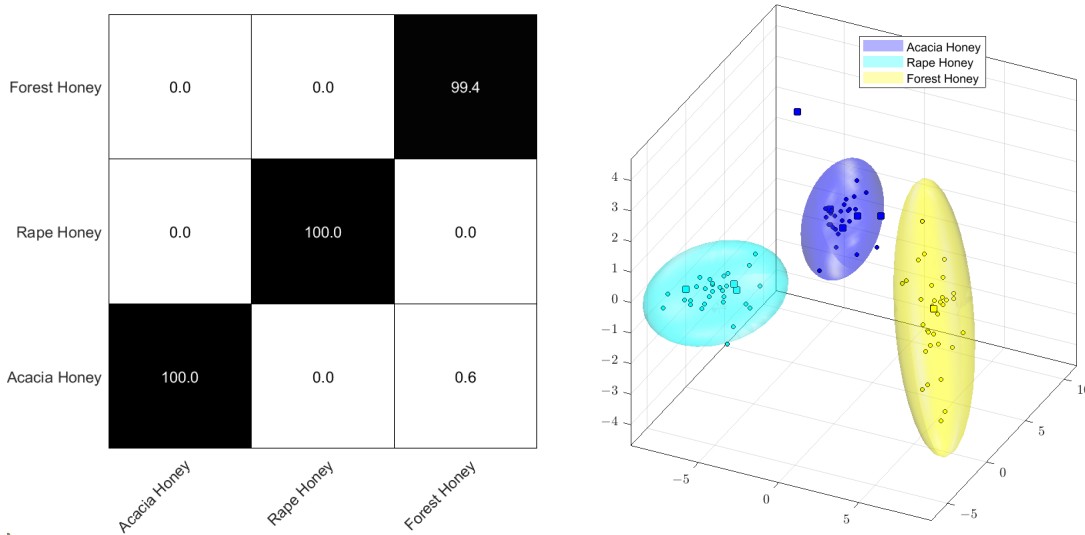

**Figure 5.** PCA−LDA Monte Carlo confusion matrix (**left**) and scores plot (**right**) of the [1]H NMR spectra of acacia, rape, and forest honey.

*5.4. Coffee*

Coffee is a beverage with an important global market share. For example, German annual coffee consumption was about 6.5 kg of coffee beans per capita [106], with the top three coffee-loving nations having annual per capita consumptions higher than 10 kg [107]. There is a growing demand for higher quality coffee with declared origin, botanical variety, and/or processing methods and the significantly higher market prices for such specialty coffees are opportunities for food fraudsters. For example, in 2018/19, OPSON VIII, an operation coordinated by INTERPOL and EUROPOL, investigated (among other foodstuffs) coffee sold on the European market [108]. This was the first official use of NMR in a legal action. With a turnover of under two days, almost 400 coffee samples were analyzed; nine were found to be diluted with cheaper *C. canephora* beans.

Gottstein et al. have recently presented NMR-based test procedures verifying several aspects of coffee: checking the declaration as "100% Arabica", authenticating different declared geographical regions of origin, checking the declaration of "organic production", and quantifying relevant ingredients [109]. Over 600 roasted coffee samples were evaluated using this procedure of authenticity, using hydrophilic and lipophilic extraction for sample preparation. In total, 400 MHz of $^1$H NMR spectra of the coffee extracts were recorded and evaluated via MATLAB™ using both targeted analysis for compounds such as 16-*O*-methylcafestol, a marker substance for *C. canephora*, and non-targeted analysis using principal components analysis-linear discriminant analysis (PCA-LDA). NMR spectral intensities were normalized relative to the signals of the respective internal standards and binned for data reduction. Spectral regions with resonances not originating from coffee metabolites (e.g., from solvent, internal standard, water) were excluded from further evaluation. Binary models were calculated for each authentication aspect (e.g., country of origin vs. other countries). From all sample scores belonging to a class, confidence ellipsoids were calculated using a 95% threshold. All PCA-LDA models were validated by an internal full ten-fold cross-validation splitting the samples from 90% to 10% (training-subset to test-subset), repeated with an outer ten-fold Monte Carlo resampling, leading to 100 validation models. The LDA assigned each sample to a class according to the lowest distance of the sample's PCA score to each class's mean. A comparison of the original class versus the assigned class was visualized with a confusion matrix, showing the average values over all 100 validation runs. Furthermore, about 20% of all samples were not used to establish the PCA-LDA models but reserved for subsequent external validation.

Along with 18 coffee compounds, including caffeine and chlorogenic acid, 16-O-methylcafestol can be quantified reliably from a limit of quantification (LOQ) of 20 mg/kg upwards. This allows the detection of dilutions of *C. arabica* with *C. canephora*, even at low levels [109].

For some of the investigated classifications, the chosen PCA-LDA may be judged as fit for the purpose of an authentication screening (Figure 6). So far, data seem to indicate that the geographical differentiation works slightly better from the hydrophilic extracts. A correct classification was possible for African coffees vs. the rest of the world and for Brazilian coffees vs. other continents, with a reliability better than nine out of ten. For Asian coffees vs. the rest of the world, for North vs. South American coffees, and for Ethiopian coffees vs. other continents, the correct classification was possible with a reliability of nine out of ten. For Colombian coffees vs. other continents, a correct classification was only possible with an insufficient reliability slightly better than eight out of ten. The differentiation according to roasting grade using the lipophilic extract correctly assigned 98% of dark coffees and 94% of light roasted coffees. The PCA-LDA of hydrophilic extracts classified 97% of drum-roasted coffees correctly and 95% of hot-air-roasted coffees. The differentiation of cultivation methods (organic vs. conventional) was insufficient, with a 71% to 74% reliability, being slightly better from lipophilic extracts [109].

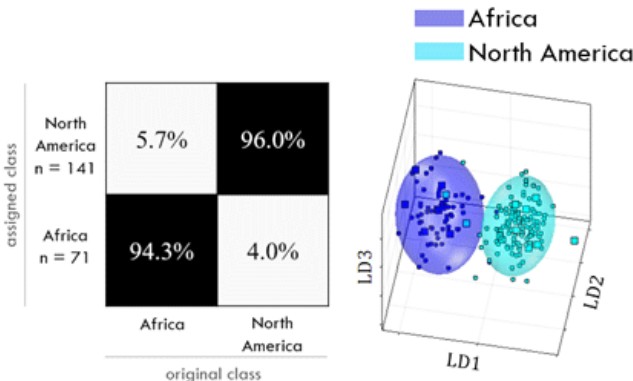

**Figure 6.** Example of a confusion matrix (**left**) and scores plot (**right**) for the differentiation of African and North American coffee samples. The confusion matrix shows the percentages of correctly and falsely assigned classes; the scores plot visualizes the variances in one class and between classes and it shows the confidence ellipsoids (95% threshold). Reprinted from Gottstein, V.; Lachenmeier, D.W.; Kuballa, T.; Bunzel. M. [1]H NMR-based approach to determine the geographical origin and cultivation method of roasted coffee. *Food Chem.* **2023**, *57*, 137278 [109], Copyright 2023, with permission from Elsevier.

## 6. Conclusions

Nuclear magnetic resonance spectroscopy is widely recognized as a versatile and innovative analytical technique, applicable to a range of fields, including food surveillance, pharmaceuticals, and cosmetics. As shown in various examples, aside from traditional methods, such as structure elucidation and the quantitative analysis of individual and multiple substances, NMR is ideal for utilizing chemometric methods to analyze spectra obtained under controlled conditions. This enables the analyst to assess authenticity, classify variations, and, for many products, establish origin. Objective assessments can be made, as spectral information is acquired in a core-specific manner. Notably, NMR analyses require often little sample preparation and measurement time. Thus, a high sample throughput and screening with a substantial information gain in the routine can be achieved, which is not provided to the same extent by any other analysis technique used thus far. Numerous methodological and technical advancements, including the employment of benchtop devices with progressively augmented field strengths, seem to be well-suited for implementing the NMR technique in routine laboratory practices [110].

As a result of the successful studies compiled above, NMR has progressed to become part of standardized applications: under Japanese project leadership, ISO published a standard for the purity determination of organic compounds used for foods and food products [6] and CEN is about to publish EN 17992 [111], standardizing the determination of the sum of 16-*O*-methylcafestol, 16-*O*-methylkahweol, and their derivatives in roasted coffee by [1]H-qNMR, thus enabling the efficient authentication of pure *C. arabica* coffee.

Since November 2021 the technical rule "Determination of several ingredients, additives, and contaminants in alcohol-free beverages by quantitative nuclear spin resonance spectrometry" (L 32.00-6) [57] is part of the German Official Collection of Methods of Analysis. In the next few years, several other standardization projects using NMR as the analysis technique will be published and will further confirm NMR as a reliable tool for routine applications.

Collaboration between several laboratories should grow in the future, combined with the factorial design of experiments. This will economically and efficiently lead to highly robust validation data and, thus, wide acceptance of NMR testing methods.

Benchtop NMR will offer new opportunities for analysis tasks of lesser complexity if limited to analyte concentrations higher than ~10 μmol/kg in the original sample material.

**Author Contributions:** Conceptualization, T.K.; methodology, T.K.; data curation, T.K.; writing—original draft preparation, T.K.; writing—review and editing, K.H.K., J.T. and D.W.L.; visualization, T.K. and K.H.K.; supervision, T.K.; project administration, T.K. All authors have read and agreed to the published version of the manuscript.

**Funding:** This research received no external funding.

**Data Availability Statement:** Not applicable.

**Acknowledgments:** The authors would like to thank NMR team of the CVUA Karlsruhe for excellent technical support and Andreas Scharinger for assistance with the illustrations.

**Conflicts of Interest:** The authors declare no conflict of interest.

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
