# Peer review of "Liquid Nuclear Magnetic Resonance (NMR) Spectroscopy in Transition—From Structure Elucidation to Multi-Analysis Method"

_separations, doi:10.3390/separations10110572_

Round 1

Reviewer 1 Report

Comments and Suggestions for Authors

The manuscript of T. Kuballa and co-workers approaches analytical procedures in NMR spectroscopy to assess the stability, quality control, and adulteration in distinct samples, such as food, cosmetic and pharmaceutical fields. This review highlights the applications of NMR spectroscopy, mainly in the quantification procedures. Considering the great number of original articles and reviews which describe the qNMR technique into the literature, or even structural elucidation, it is strongly important to mention newer devices and protocols which were incorporated in the last decades and/or to bring historical aspects of the development of NMR. Otherwise, the manuscript will only describe simple examples of application of NMR and qNMR, do not mentioning the progress and the state-of-the-art. Thus, the review proposal is not clear and I have some suggestions which can be useful to improve the discussion and, specially, giving more details of the qNMR technique for the readers.

suggestions:

- Introduction: In this section, I believe that is important to clarify adding the NMR equation, which is used to quantify analytes (great part of the manuscript is related to qNMR).

- Sample Preparation: In this topic the authors should mention the advantages and disadvantages of NMR considering the following aspects: Advantages: non-destructive nature, no need derivation, simple and rapid preparation, and capability to evaluate complex mixtures (no need isolation and purification). Additionally, to compare the disadivantages of other techniques, as chromatography and/or mass spectrometry.

- NMR spectroscopy: In this topic the authors should mention other nuclides, as 31P and 19F, and describe the aspects that affect the resolution and quantitative accuracy of qNMR (Maybe brings some statistical aspects). In all themes, review the limitations and applications.

- Structure Elucidation and Confirmation: Along with the pharmaceutical examples, the authors should mention some agrochemical evaluation, which is used for structure elucidation and formulation evaluation. In addition to this discussion, please, give some examples of newer NMR experiments (such as "pure-shift") to visualize the recent improvements in this topic.

- Quantification and Purity Check for Quality Management: I recommend the authors to review the evolution of NMR pulse sequences, focusing on the water suppression, strongly important in the quantification of food analysis. In this topic could be mentioned the devices used to standardization of NMR samples, as auto-sampler and qNMR calibrants (give some examples along with limitations).

- Chemometric Methods - Food Fraud and Authenticity Assessment: I suggest to reference others authors to exemplify the applications of qNMR in food, once there are 22 references of Kuballa (35%). Along with the own results, the authors can bring more references for the same examples, which you be more useful to see the literature, and to compare goals and results.

- I recommend to add some recent reviews regarding qNMR spectroscopy: https://doi.org/10.1002/mrc.5099; https://doi.org/10.1021/acs.analchem.1c02056; https://doi.org/10.3390/magnetochemistry7010015; 10.1088/1681-7575/ab336b; https://doi.org/10.1007/s11306-015-0794-7; 10.2174/9789811439971120080003; 10.2174/9781681081434116040007; https://doi.org/10.3390/molecules22020247; https://doi.org/10.1155/2022/7490691; https://doi.org/10.1590/fst.43622; https://doi.org/10.3389/fmolb.2022.882487; https://doi.org/10.1021/acs.analchem.2c04606.

Based on these reviews, the authors could provide some thoughts about the challenges that face the qNMR spectroscopy and/or the next steps to improve the qNMR techniques.

Author Response

Thank you for considering our script for peer reviewing, we greatly appreciate your comments and suggestions. The answers are listed with a yellow background in the attached word document for the respective comment or proposal.

Best regards

Reviewer 2 Report

Comments and Suggestions for Authors

In this manuscript, the author attempted to review and summarize the application of NMR analysis technology based on multivariate statistics in the fields of food, cosmetics, pharmaceuticals, etc., and listed many practical cases he has studied before.

However, there are many issues that need to be addressed.

1.      In the Abstract and Introduction, the author emphasized that the purpose of the review is the application of NMR technology in the food, cosmetics, and pharmaceutical industries. However, the main content of the title and manuscript is related to multivariate statistical analysis in NMR, which is only a part of the practical application of NMR. Please further clarify the purpose of this review.

2.      Based on the author's cases, it can be seen that the author has already done a lot of research work in the field of food. However, there are also many other outstanding research works in the fields of drugs, food, and cosmetics, some of which could also be used as case studies.

3.      The title of the manuscript is the transformation from structural elucidation to multivariate statistical analysis. One of the main purposes of multivariate statistical analysis is to identify the chemical components (or metabolites) that cause differences, based on the identification of chemical components (or metabolites). Therefore, the work of structural elucidation is very important. Please introduce how to perform structural elucidation when giving examples.

4.      The author has introduced a lot of research on food, but the research on drugs, cosmetics, etc. is not enough. Please continue to supplement.

5.      There are many methods for multivariate statistical analysis. Please supplement the relevant research progress and not limit it to PCA-LDA analysis examples. Software is not just "SIMCA", but also supplements relevant content.

6.      After careful examination of the literatures, it was found that the examples and some spectra cited by the author did not appear in the cited literatures. The review should be based on published literatures and data for summary and analysis, and should not be a secondary processing of data, especially for the reprocessing of raw data that is not publicly available.

7.      The preparation of samples is very important, but the manuscript is very simple. Please supplement.

8.      Please provide quantitative parameter information and main NMR experiments in the NMR experiment section. Additionally, “segment” is not the only type of data used for statistical analysis.

9.      Line 61, “2H-NMR spectroscopy”, It's very strange, Please check.

10.    Line 120-121, “preservation” can confuse beginners. Please explain in detail.

11.    Line 146-156, in many cases, NMR is not the preferred detection item, such as in the identification items in the pharmacopoeia. HPLC-UV, HPLC-MS, GC-MS are common methods for substance identification.

12.    Line 162-163, the impurity signals of Losartan may overlap with that of Losartan, so it cannot be determined that there are no impurities based solely on a 1H NMR.

13.    Figure 1 The chemical shifts and integral areas in the spectrum should be indicated.

14.    It should be noted that chemical shifts do not have units. Please do not use “ppm” as it is not an international unit.

15.    What is the detailed content of Line 173, "The instrument parameters and criteria of ISO IEC 17025"?

16.    Line 174, "a standard procedure" is only a research method and cannot become a standard detection method. There are also doubts about whether the detection capabilities of some components in this literature meet the requirements of relevant standards.

17.    The literature [34] highlights the advantages of benchtop NMR, while this article reviews high-resolution NMR. Reference [35] is a summary of the meeting, not a research paper.

18.    Line 178-181, the content of 'PULCON' is too simple, and references [22,34-37] seem to be unrelated to PULCON.

19.    19. Line 189-192, Reference [38] did not use the NMR method, and it showed that samples from Slovenia and Australia had the highest quality issues (>10 issues), while Russia had zero issue. Please check if the description of '304 spirits' is accurate.

20.    20. Figure 3 PCA-LDA scores plot (left) is inconsistent with the description in the original literature.

21.    Line 201-202, is there a clear restriction requirement in the industry standard for "Higher aldehydes are critical indicators of autonomous fat spill"?

22.    Line 213-214, “the PLS method is often employed for 213 screening NMR analyses of food and beverages”, What is confusing is what is used for screening?

23.    Line 232-235, experimental information is inconsistent with the cited literature.

24.    5.2. Edible Oils, the content in the manuscript is inconsistent with reference [53].

25.    Line 300-307, experimental information is inconsistent with the cited literature [58].

Figure 5. Is it the original image of the literature or a reprocessing based on the original data?

Comments on the Quality of English Language

Further modifications needed

Author Response

(The authors gave the same response as above.)

Reviewer 3 Report

Comments and Suggestions for Authors

The article written by Thomas Kuballa et al. concerns the increasingly popular NMR method in chemical, pharmaceutical and food analysis.

The work is written very well. It presents a broad view, discusses some problems, and indicates modern statistical methods supporting the analysis.

As a reviewer, I only have a few comments that I think may help to improve the quality of the (already good) article.

1. Is the article only about NMR in solution? If so, it's good to include it in the title, if not, solid state NMR requires discussion. Both in the context of sample preparation and application (especially interesting in the pharmaceutical context).

2. Line 149 - mass spectrometry better reflects the ideas of this method than mass spectroscopy, the first form is preferably used.

3. Line 163 "rule out impurities" - it is worth mentioning the limited detection and limitations of the NMR method, especially in the context of detecting trace amounts of impurities.

4. In the case of losartan, it is also worth showing the 13C spectrum with its interpretation, and even the nitrogen spectrum, if it has been recorded.

5. In the context of structure interpretation, it is worth mentioning briefly the DFT computational methods that enable the calculation of shielding constants, which facilitates the interpretation of NMR spectra.

6. The summary may include brief information about future prospects. Especially in the context of the popularization and development of benchtop spectrometers.

Comments on the Quality of English Language

Minor editing of English language required

Author Response

(The authors gave the same response as above.)

Reviewer 4 Report

Comments and Suggestions for Authors

The review by Kuballa et al concerns with a very interesting topic regarding applications of NMR spectroscopy on foods, cosmetics and pharmaceuticals. it is well written with food for thoughts, yet the bulk of rhe references is extremely old, with less than ten of sixttytwo were less than 5 years old.  therefore I suggest the authors to add more recent references such as :

1.       NMR-Based Approaches in the Study of Foods

by Anatoly P. Sobolev 1ORCID,Cinzia Ingallina 2ORCID,Mattia Spano 2ORCID,Giacomo Di Matteo 2ORCID andLuisa Mannina 2,*Molecules 2022, 27(22), 7906; https://doi.org/10.3390/molecules27227906

 2.       ,The use of time domain 1H NMR to study proton dynamics in starch-rich foods: A review, Isabella M. Riley, Mieke A. Nivelle, Nand Ooms, Jan A. Delcour, COMPREHENSIVE REVIEWS IN FOOD SCIENCE AND FOOD SAFETY https://doi.org/10.1111/1541-4337.13029

 3.       Applications of nuclear magnetic resonance spectroscopy to the evaluation of complex food constituents, Ruge Cao a b, Xinru Liu b, Yuqian Liu b, Xuqing Zhai b, Tianya Cao c, Aili Wang d, Ju Qiu e, Review Food Chemistry,

 4.       Nuclear Magnetic Resonance (NMR) Spectroscopy in Food Science: A Comprehensive Review: NMR spectroscopy in food science, Comprehensive Reviews in Food Science and Food Safety , Emmanuel Hatzakis 18(3)DOI:10.1111/1541-4337.12408

 5.       Application of nuclear magnetic resonance in food analysis,Review Article • Food Sci. Technol 42 • 2022 • https://doi.org/10.1590/fst.43622  

 6.       H-NMR metabolomics for wine screening and analysis, Inès Le Mao, Grégory Da Costa, Tristan Richard,Vol. 57 No. 1 (2023): OENO One, DOI: https://doi.org/10.20870/oeno-one.2023.57.1.7134

 7.       Application of nuclear magnetic resonance in food analysis,Qian QU1, , Lan JIN1, Food Science and Technology,DOI: https://doi.org/10.1590/fst.43622

Author Response

(The authors gave the same response as above.)

Round 2

Reviewer 1 Report

Comments and Suggestions for Authors

The authors have performed the changes into the main text according to the suggestions.

Author Response

In view of your comment, we assume that you recommend the manuscript for publication in its current form. Thank you again for reviewing our manuscript. We value your feedback and appreciate your recommendation for publication.

Best regards

Reviewer 2 Report

Comments and Suggestions for Authors

1.      In the Abstract (Lines 21-23) the author emphasized that the purpose of the review is the application of NMR technology in the food, cosmetics, which is inconsistent with the main idea of the manuscript. Please further clarify the purpose of this review.

2.      The author replied that adding research cases from others would exceed the manuscript. Please note that the review is not a copy and paste of previous research results, but rather a summary of the patterns of many previous research results. Please continue to summarize and refine the literature and provide constructive conclusions.

3.      There are many methods for multivariate statistical analysis. Although the author has claimed to have added literatures, it is still insufficient for the theme of this manuscript. It is recommended to add a section specifically discussing the methods of multivariate statistical analysis in the manuscript.

4.      After careful examination of the literatures, it was found that the examples and some spectra cited by the author did not appear in the cited literatures. The review should be based on published literatures and data for summary and analysis, and should not be a secondary processing of data, especially for the reprocessing of raw data that is not publicly available.

5.      I noticed that the author replied that some examples and spectra in this manuscript are original and have not been published before. Please note that this manuscript is a review and not a research article. Therefore, please do not use unpublished data while citing literatures, which is different from the content of the cited literature. And we are also unable to verify the data.

6.      Sample preparation is very important, but the manuscript is very simple. Although the author replied that the content would exceed that of this manuscript. However, please note that the overview is not simply a copy and paste, but a summary and refinement. Please summarize and supplement by the author.

7.      Please provide quantitative parameter information and main NMR experiments in the NMR experiment section. Additionally, “segment” is not the only type of data used for statistical analysis. Please summarize and refine the manuscript before supplementing it, so as not to exceed the word limit of the manuscript.

8.      Line 151, “2H-NMR spectroscopy”, It's strange why the author still cited a literature [15] that did not use NMR technology after revising it again?.

9.      Lines 286-288, Although the author has made modifications. However, relying solely on 1H NMR still cannot confirm purity. If the signals of impurities overlaps with the signals of the sample, it is impossible to determine purity or exclude impurities.

10.    Lines 284-286, Please explain what software the author used and under what solvent conditions the data was predicted. And the predicted chemical shifts shown in Figure 1 are inconsistent with the measured data, which is inconsistent with the conclusion described in the manuscript. For example, the chemical shifts of multiple signals in the aromatic region are inconsistent.

11.    It should be noted that chemical shifts do not have units. Please do not use “ppm” as it is not an international unit. It is true that "ppm" was commonly used in some early published books, but currently some mainstream authoritative journals such as Journal of Natural Products (ACS) have begun to correct this incorrect usage. As this is an introductory article, the wording should be more rigorous. And currently,  the correct way of expressing chemical shifts has been accepted by many chemical researchers.

12.    Lines 329-330, The statement 'established a standard procedure' is not rigorous enough. As the author replied to 'Further more', according to the cited literature [49], there is no detailed description similar to SOP, nor is it concluded that they have been validated and applied in multiple laboratories. If so, the author should provide more application information.

13.    Lines 202-204, reference [34] emphasizes the advantages of desktop NMR, but desktop NMR is not high-resolution NMR. Please note that this review is a summary of high-resolution NMR, as described by the author in Line 190 ' high resolution NMR is applicable in the quality control and surveillance '.

14.    The reference [35] is a conference abstract, not a research paper. The authenticity of this conclusion has not been peer reviewed, therefore it is not appropriate as supporting evidence.

Author Response

Thank you for considering our script for peer reviewing, we greatly appreciate your comments and suggestions. The answers for your respective comments or proposals are listed with a yellow background in the attached word document.

Best regards

Reviewer 3 Report

Comments and Suggestions for Authors

The Authors have improved the aticle. In my opinion it can be accepted now in Separations.

Comments on the Quality of English Language

English is fine.

Author Response

Thank you again for reviewing our manuscript. We value your feedback and appreciate your recommendation for publication.

Best regards